# Archaeobotanical and chemical investigations on wine amphorae from San Felice Circeo (Italy) shed light on grape beverages at the Roman time

**Louise Chassouant**[1,2]*, **Alessandra Celant**[2], **Chiara Delpino**[3], **Federico Di Rita**[2], **Cathy Vieillescazes**[1], **Carole Mathe**[1], **Donatella Magri**[2]*

**1** IMBE UMR7263 / IRD237, Avignon University/ CNRS/ IRD/ AMU, Restoration Engineering of Natural and Cultural Heritage, Avignon, France, **2** Dipartimento di Biologia Ambientale, Sapienza Università di Roma, Roma, Italy, **3** Soprintendenza ABAP per le Province di Frosinone e Latina, Rome, Italy

* louise.chassouant@uniroma1.it (LC); donatella.magri@uniroma1.it (DM)

## Abstract

We hereby investigate the pitch used for coating three Roman amphorae from San Felice Circeo (Italy) through a multidisciplinary study. The identification of molecular biomarkers by gas chromatography—mass spectrometry is combined with archaeobotanical evidence of pollen and plant tissues of *Vitis* flowers. Diterpenic chemical markers together with *Pinus* pollen and wood revealed Pinaceae tar coating. Aporate 3-zonocolpate pollen, identified as *Vitis*, together with tartaric, malic and pyruvic acids elucidate the grape-fermented nature of the content. Our conclusions open new consideration on the use of grape derivatives that cannot be supported by traditional analytical methods. Based on the finds of aporate *Vitis* pollen, found also in local modern and Middle Pleistocene samples, we hypothesize the use of autochthonous vines. The presence of a medicinal wine (historically reported as *oenanthium*) is also considered. We interrogate *Vitis* pollen capacity to target grapevine domestication, thereby providing innovative tools to understand such an important process. We anticipate our study to encourage a more systematic multidisciplinary approach regarding the analyses of wine amphorae.

## Introduction

Pilot experimental protocols with the specific aim of accessing pollen trapped in organic resins of archaeological artefacts were advanced by Pons [1], Arobba [2], Jones *et al.* [3], and Jacobsen and Bryant [4]. From there, only a limited number of pollen studies have been conducted on amphorae. They mainly focus on the liquid recovered from sealed jars [5, 6] and on sediments contained in the ceramics from cargo containers from marine contexts, with the objective of identifying pollen or phytoliths [7]. Pollen analyses from resins of archaeological artefacts has been little used with the purpose of better understanding the history beyond the object. Significant *Pinus* and *Vitis* pollen representation highlighted pine pitch coating used

**Data Availability Statement:** All the data related to pollen and chemical analysis are within the main text of the paper (in Table 1 and the figures).

Archaeobotanical pictures (pollen and vegetal tissues) are shown in the figures. Chemical information is shown in the chromatogram in the figures and the main text. The Supporting Information file includes two pictures of SEM microscopy showing morphological features of Vitis already described in the main text. No other data were collected.

**Funding:** This research was financially supported by Marie Skłodowska-Curie Innovative Training Network, The ED-ARCHMAT European Joint Doctorate, H2020-MSCA-ITN-EJD ED-ARCHMAT Joint Doctorate (LC Project ESR9, grant agreement no 766311). The funders had no role in study design, data collection and analysis, decision to publish, or preparation of the manuscript.

**Competing interests:** The authors have declared that no competing interests exist.

for wine jars [2, 8–11]. Coupling palaeobotanical to isotopic and chemical characterization, Arobba *et al.* [6] were able to trace back the oenological content of sealed amphorae from a Roman shipwreck, as well as the central-southern Italian provenance of the cargo. Although they demonstrated the effectiveness of pollen analyses in the identification and characterization of the nature and geographical origin of the transported wine, their methodologies have barely been followed and similar investigations are still rare. Even other types of organic materials, e.g., rope, caulking material, laces watercrafts, and organic coffin have been seldom investigated through pollen [12–16]. Archaeobotany has been often combined with other analytical disciplines to promote interdisciplinary approaches [17–23] but palynology is still barely associated to chemical analyses [11, 24–26].

At the same time, analytical methods are increasingly interested in using cutting edge techniques applied to archaeological materials. Among them, liquid or gas chromatography coupled with mass spectrometry (GC-MS) dominate the field, due to highly sensitive and selective capacities to target molecules [21, 25, 27–29]. Retention time and molecular fragmentation account for trustworthy molecular identifications [30]. Scientific and archaeological consensus are prevailing, herewith stating that independent evidence must be sustained to assess with certainty the history of the containers [31]. Organic analyses residues aims at extracting and interpreting molecular markers either trapped in the organic coating or in the potsherd matrix of vessels [32]. Archaeological interpretation naturally derives from the absence and/or presence of such biomolecular indicators [33]. However, caution is needed when interpreting chemical analyses in archaeological terms. Regarding the possible overinterpretation due to the presence of the tartaric acid in chemical analyses, up to now considered as a grape marker [34]. The resort of control becomes indispensable since chemical analyses cannot support archaeological interpretation on its own. Indeed, tartaric acid can be released from phthalates contained in plastic bags under acidic treatment [31], and it can also migrate from surrounding soils [35]. Systematic sampling and analyses of sediments associated to the studied materials are highly recommended to prevent false positives [25]. However, the awareness of this problem is recent, and the question remains open for artefacts excavated long time ago, washed, restored and preserved in deposits and museums, for which no associated sediment is available.

In the present work, three marine amphorae, retrieved in 2018 from the ancient anchorage of San Felice Circeo (Italy), offered a rare opportunity to develop interdisciplinary research through archaeobotanical and chemical analyses. The aim of this article is to discuss the effectiveness of a multidisciplinary approach, initially developed to identify the nature of the organic content of the amphorae, to trace back the history beyond the artefacts.

## Materials and methods

### Archaeological materials

**Archaeological context.** In 2018 notable winter storm tides have allowed to identify a huge scattering of different archaeological finds on a seabed close to the modern harbor of San Felice Circeo (Latina–Italy), ca. 90 km SE of Rome (41˚13'49.0"N, 13˚06'30.1"E). The area is located at a distance of about 500 m from the present coastline; the depth of the seabed varies from 5 to 7 m under the sea level.

Since then, regular underwater archaeological surveys have been conducted by the Soprintendenza Archeologia Belle Arti e Paesaggio per le province di Frosinone e Latina (the local Office of the Italian Ministry of Culture) in order to elaborate a seabed mapping of the archaeological area, to delimit the zones of sherd scattering, and to obtain a clearer framework of underwater record. These surveys, which are still ongoing, revealed a broadly consistent

chronological representation, with ceramic finds spanning from the Republican period through the Late Roman period up to the post medieval period. The limited amount of morphologically and chronologically similar ceramic containers, the fragmentary state of most of the recovered pots, and the pattern of dispersion may be interpreted as an evidence of an ancient anchorage area (Delpino and Melandri 2018, unpublished data). In previous topographic studies, the existence of a Roman port close to the finding area was supposed mainly because of the presence of the hypothesized ancient mouth of the *Fossa Augusta* (a hydraulic canalization which was probably conceived in I century A.C. and then attributed to Nero) and because of the presence of some romans docks on the shoreline and submerged, that nowday are no longer visible [36]. As a working hypothesis, the recent discovery of various late Greco-Italic/transitional Dressel 1A amphorae also suggests the possible presence of a small shipwreck, which needs to be confirmed by underwater surveys. Hereby, 3 amphorae labelled SFC1, SFC2 and SFC5 have been studied (Fig 1). After excavation, archaeological specimens are placed at the Soprintendenza office at municipality of San Felice Circeo. No permits were required for the described study, which complied with all relevant regulations. The majority of the recovered Roman amphorae belongs to late Greco-Italic (referred to as Lyding Will e) and Dressel 1A type, dating from the second half of 2nd century BC to the middle of the 1st century

| Sample label | Description | Amphorae |
|---|---|---|
| SFC1 (20.S321-31.884) | Late Greco-Italic amphora / Dressel 1A amphora. Painted inscription (*titulus pictus*). Pronounced triangular rim, high cylindrical neck, lightly narrowed at the bottom and thin, s-shaped handles, rounded shoulder. The label L. M. is difficult to interpret. Second half of the 2nd century BC. |  |
| SFC2 (20.S321-31.858) | Dressel 1A amphora. Cylindrical body shape with an angular shoulder. The bottom of the neck is cylindrical; the upper part of the neck and handles are not preserved. Last quarter of the 2nd century - first half of the 1st century BC. |  |
| SFC5 (20.S321-31.875) | Lamboglia 2 amphora. Thick-walled bag-shaped body; neck and handles are not preserved; the spike is broken. |  |

Fig 1. Investigated archaeological amphorae.

BC. The late Greco-Italic type is a wine amphora with a wide distribution in the Mediterranean from the second quarter of $2^{nd}$ century up to around 140–130 BC. The latter Italic Dressel 1A amphora is an evolution of the late Greco-Italic type [37]. The transition to the new container is not sudden and does not involve a clear break with the previous production; the transitional type is known as "Lyding Will e". Dressel 1A, the most common among late Republican Roman amphorae, were mostly filled with wine [38, 39]. Mainly produced in southern-central Italy, from Campania to Etruria where a number of kiln sites along the coastal area are known, these amphorae have widely circulated in Gaul, Britain, Spain and central Europe [38–40]. The manufacture area does not necessarily coincide with the loading site. However, considering the possibility of San Felice to be a center of redistribution and assuming the presence of a manufacture site nearby [41], we can hypothesize that the loading site was San Felice itself, highlighting a production site in Latium for the studied amphorae SFC1 and SFC2 (Fig 1).

The third investigated amphora SFC 5 belongs to Lamboglia 2 amphorae (Fig 1). Coming from the Adriatic coast [39, 42], Lamboglia 2 were widely distributed throughout the western Mediterranean but a production in western Italy alongside the Dressel 1 amphorae has also been suggested [42]. This typology was meant for the maritime transport of wine or olive oil [43, 44]. The analyses of the vessels from the Madrague de Giens shipwreck suggested wine content [45]. Wine is strongly suggested regarding the remaining presence of internal resin coatings observable in numerous Lamboglia 2 [39] found in different shipwrecks such as Cava-liére A (n˚ 282), Cap Roux n˚ 197), Punta de Algas (n˚ 9191), Ponza (n˚ 1060) [46]. The olive oil hypothesis is disfavored as it would have reacted with the pitch, degrading the oil quality and taste [47].

## Chemical analyses

**Gas chromatography–Mass spectrometry: Sample preparation and equipment.** Samples of 20 mg of archaeological coatings were recovered from the internal body and bottom of the amphorae by scraping the organic layer with a scalpel and were treated following a two-step protocol [29]. The first extraction corresponds to an organic lipid extraction (labelled 1LE) while the second step is a microwave-assisted transesterification catalyzed by a Lewis acid (2LE-MW).

The extracted samples were trimethylsilylated and dissolved in 0.2–0.6 mL of hexane/DCM (1/1, v/v) before injection. GC–MS analyses were performed on a Thermo Scientific™ Focus system equipped with a Thermo Scientific Al 3000 autosampler and coupled to a Thermo Fisher Scientific™ ITQ™ 700 Series Ion Trap Mass Spectrometer. The separation was achieved on a 30 m x 0.25 mm internal diameter x 0.25 μm film thickness fused silica capillary column Thermo-GOLD™ TG-5MS (5% diphenyl; 95% dimethyl polysiloxane). 1 μL solution was injected in split-less mode at 250˚C. The transfer line and the ion trap were respectively maintained at 300 and 200˚C. Molecular components were carried by a constant 1 mL/min helium flow. Data treatment were carried out on Xcalibur software. Molecular compounds were identified by retention time, comparison with mass spectrum of commercial molecular standards, with the internal molecular library of the laboratory and with NIST MS Search 2.0 database recorded with an electronic ionization of 70 eV. The oven temperature was held at 50˚C for 2 min, increased to 140˚C at 8˚C/min held for 2 min before reaching 160˚C at 2.5˚C/min and finally 330˚C at 15˚C/min and held for 3 min. Spectra were recorded in the 50–650 $m/z$ mass range.

## Archaeobotany

Reference modern wild grapevine flowers, both male and female, as well as fruits, used in this study were collected near Rome, in the municipality of Morlupo (42˚09'19"N; 12˚30'26"E).

Reference pollen grains were also collected from the surface of the fruits of a wild grape from Tivoli (41˚57'09"N; 12˚49'04"E). All grapes were sampled in wooded rims of river valleys, within a riparian vegetation characterized by *Quercus cerris*, *Q. pubescens*, *Fraxinus ornus*, *Ulmus minor*, *Populus nigra*, and *Alnus glutinosa*.

As a reference for pre-domestication *Vitis* pollen, the Middle Pleistocene diatomite sediments from Fosso di San Martino [48], located in the municipality of Rignano Flaminio (42˚ 11'26"N, 12˚31'13"E), near Rome, were re-analysed to observe the morphological characters of wild pollen grains in the region.

Adapted from [12], pitch samples of ca. 0.5 g were systematically dissolved in tetrahydrofuran and ethanol before the addition of a tablet with a known number of exotic *Lycopodium* spores to estimate the pollen concentration. To limit contamination from modern pollen grains, whole pieces of pitch were treated. Acetolysis was not needed.

Modern pollen was acetolyzed following the standard procedure [49]. Modern fruits from *Vitis vinifera* subsp. *sylvestris* were hydrated in water for 12 hours before being heated for 10 min in NaOH (10%) and acetolyzed.

Pollen was observed under a Zeiss Axioscope light microscope at 400x and 630x magnifications. Identifications were supported by pollen morphology atlases [50–52]; websites https://www.paldat.org; https://globalpollenproject.org, and the reference collection of the Laboratory of Palaeobotany and Palynology of Sapienza University of Rome. Morphological pollen and wood observations were also performed by Environmental Scanning Electron Microscope (ESEM) Hitachi TM-3000 Tabletop operating at 15Kv without previous coating. The images were recorded at magnifications varying from 150x to 700x.

## Results

### Chemical analyses

For all the amphorae SFC1, SFC2 and SFC5, the first extraction (1LE) with polar organic solvents (DCM:MeOH) interestingly revealed a conifer resin made out of Pinaceae wood tar (dehydroabietic acid (DHA); methyldehydroabietate (DHAM)) (Fig 2A). Aromatized (retene, norabietatrienes) and oxidized (hydroxy- and oxo-DHAM derivatives) abietanes highlighted a high temperature formulation and the ageing of the resin [53, 54]. Oxidized abietanes were identified through their characteristic fragment ions ($m/z$ 191; 253) [55].

All the archaeological coatings contained tartaric ($m/z$ 276), malic ($m/z$ 303) and pyruvic ($m/z$ 61, 89, 117, 173) acids, greatly identified as butylated and butylacetal derivatives. Syringic acid was only identified in the amphora SFC1 and never present in both extractions for SFC2 and SFC5. For all the amphorae, succinic and glutaric acids were identified in 1LE through their characteristic fragment ion $m/z$ 147 in the trimethylsilylated form while no traces could be identified in 2LE-MW.

Since acids were transesterified in the second step, molecules identification in 2LE-MW was restricted to grape derivatives markers (succinic, pyruvic, malic, tartaric and syringic acids). Considering the presence of diethyl or butyl ethyl grape acids reported by Garnier and Valamoti [21] in a Neolithic jar, similar reactions were controlled in our samples but no esters were observed. Such compounds would indeed be produced by esterification with the ethanol contained in the fermented beverage.

### Archaeobotanical analyses

**Pollen.** The pollen concentration of samples SFC1, SFC2 and SFC5 is noticeably low (1771, 175 and 213 pollen grains/g resin, respectively; Table 1). The number of identified pollen grains are 150, 61 and 48 respectively. SFC1 displayed the richest pollen content with a

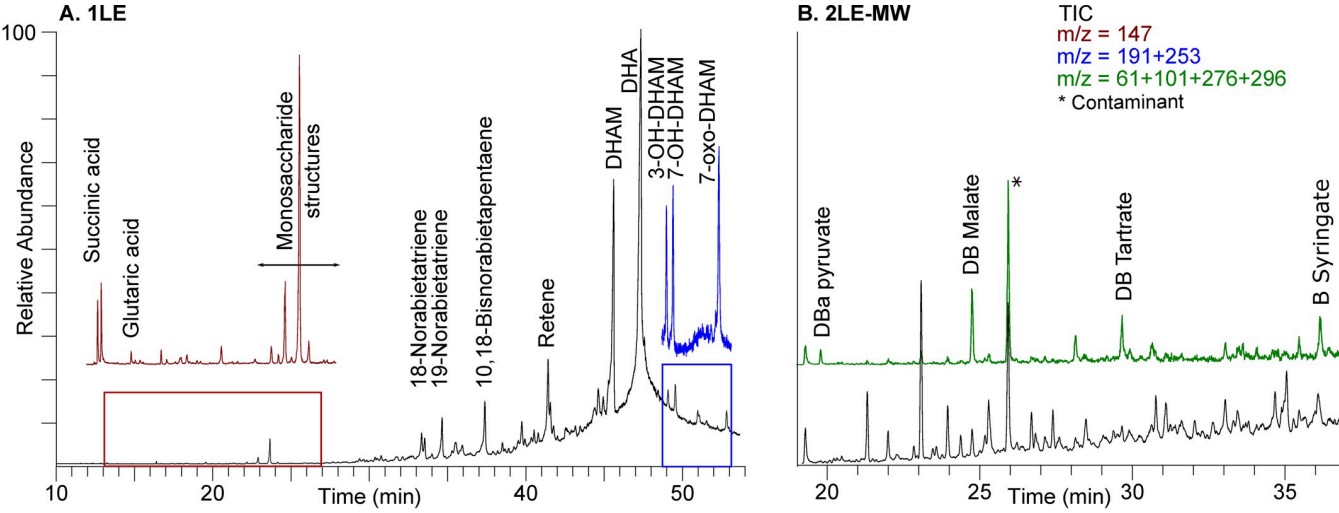

**Fig 2. GC-MS chromatogram for the pitch of SFC1.** A. represents the first extraction (1LE) and B. corresponds to 2LE-MW, the butylated second extraction. Total Ion Current (TIC) is black-colored. Red, blue and green colors refer to fragment ions searching (respectively *m/z* 147 (succinic and glutaric acids) and *m/z* 191; 253 (hydroxy- and oxo-abietanes) in 1LE and *m/z* 61; 101; 276; 296 (dibutylacetal pyruvate; dibutyl malate; dibutyl tartrate; butyl syringate in 2LE-MW).

major presence of *Quercus* (46%), *Pinus* (28.7%) and minor contributions of *Olea* (6.7%), *Phillyrea* (6%), Brassicaceae (2%), *Carpinus* and *Erica* (1.3%), *Myrtus*, *Plantago* and *Ranunculus* (0.66%). SFC2 showed a significant presence of *Pinus* (65.6%), followed by *Quercus* (16.4%), *Phillyrea* (3.3%), *Cedrus* (3.3%), and *Erica* (1.6%). SFC5 exhibited *Pinus* (35.4%) and *Quercus* (29.1%) followed by *Artemisia* and *Phillyrea* (8.3%), *Ranunculus* (6.3%), *Alnus*, *Erica* and *Ostrya* type (2.1%). *Vitis* represented 4.7%, 9.8% and 6.3% of total pollen, respectively (Table 1).

Assumption of highland pine species is suggested regarding the small pollen size (55 to 80 μm). Following the classification by Desprat *et al.* [56] and keeping in mind that fossilization can alter the grain size, the identification was possible up to the subsection including *P. mugo*, *P. nigra* and *P. sylvestris*. Unfortunately, it remained unreliable to identify the pollen grains to the species level.

In all the three pitch samples, pollen observations featured the presence of aporate 3-zonocolpate grains, ranging 20–25 μm, with psilate to micro-scabrate ornamentation (Fig 3). They displayed narrow and long slit-like colpi, making the pollen round to slightly oval in equatorial view (EV) and obtuse triangular to hexagonal outline in polar view (PV) with straight to softly

**Table 1. Pollen grains recovered from the analyses of amphorae SFC1, SFC2 and SFC5.**

| | Asteroideae | Betulaceae | Brassicaceae | Ericaceae | Fagaceae | Myrtaceae | Oleaceae | Pinaceae | Plantaginaceae | Ranunculaceae | Vitaceae | Total pollen grains | Pollen concentration (pollen/g) |
|---|---|---|---|---|---|---|---|---|---|---|---|---|---|
| SFC1 | | *Carpinus betulus* 1.3% | Brassicaceae 2.0% | *Erica* 1.3% | *Castanea* 0.66% *Quercus* 46% | *Myrtus* 0.66% | *Olea* 6.7% *Phillyrea* 6.0% | *Pinus* 28.7% | *Plantago* 0.66% | *Ranunculus* 0.66% | *Vitis* 4.7% | 150 | 1771 |
| SFC2 | | | | *Erica* 1.6% | *Quercus* 16.4% | | *Phillyrea* 3.3% | *Pinus* 65.6% *Cedrus* 3.3% | | | *Vitis* 9.8% | 61 | 175 |
| SFC5 | *Artemisia* 8.3% | *Alnus* 2.1% *Ostrya* type 2.1% | | *Erica* 2.1% | *Quercus* 29.1% | | *Phillyrea* 8,3% | *Pinus* 35.4% | | *Ranunculus* 6.3% | *Vitis* 6.3% | 48 | 212 |

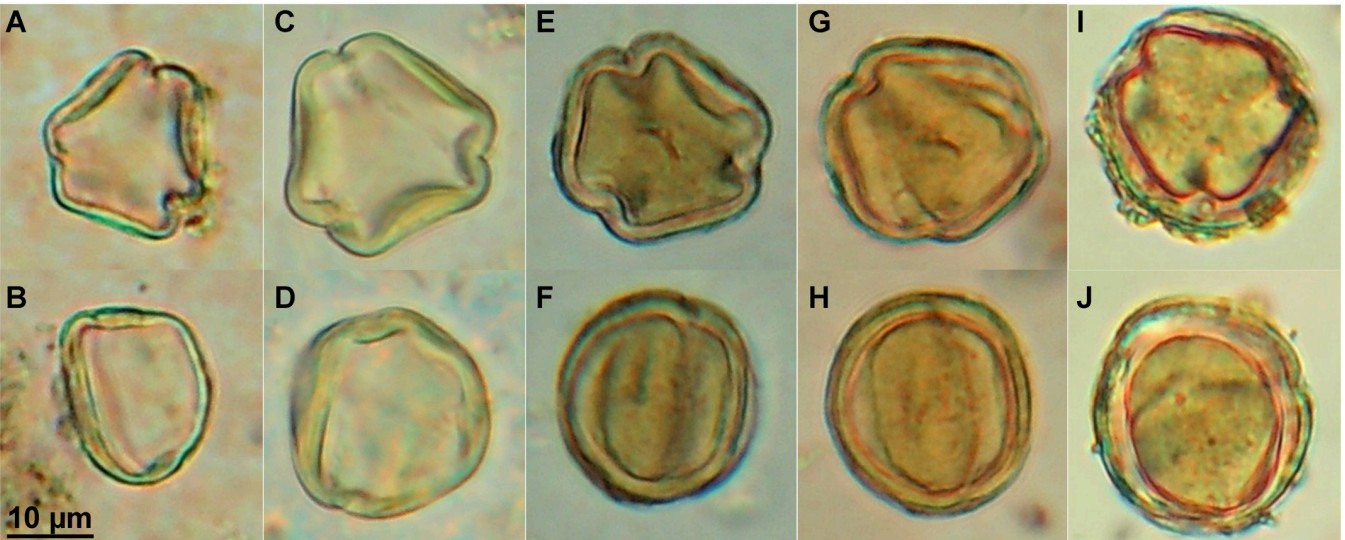

**Fig 3. *Vitis* pollen in polar and equatorial view.** Pollen grains recovered from: A, B. Fossil sediments from Rignano Flaminio (18–22 μm); C, D. Surface of modern wild fruits of *Vitis* from Tivoli (20–24 μm); E, F. Pitch of amphora SFC1; G, H. Pitch of amphora SFC5; I, J. Pitch of amphora SFC2.

concave sides. As illustrated in Fig 3, the main feature remained the total absence of porus within the colpi and a thick exine (up to almost 5 μm for the pollen grain in Fig 3I and 3J, with small residues stuck to the exine).

Pollen grains from female flowers and from the surface of the fruits of modern wild grapevines from Morlupo and Tivoli are morphologically consistent with the fossil grains from the pitch, being tricolpate, aporate and with a relatively thick wall of up to 4 μm in polar view (Fig 3C and 3D and S1 Fig). Such thickness is consistent with pollen of Balkan indigenous Žilavka and Blatina cultivars described in the literature [57, 58]. Mercuri *et al.* [59] also reported thicker exine dimension of 1.6 μm (± 0.70) in polar view for wild dioecious plants, while current "ancient cultivars" of Lambrusco Grasparossa or Bianca di Poviglio measured less than 1 μm. Likewise, although with a slightly smaller grain size (18–22 μm), the aporate 3-zonocolpate morphology of *Vitis vinifera* was also found in pollen grains from sediments belonging to the Middle Pleistocene sediments of Rignano Flaminio (Fig 3A and 3B and S1B Fig). Apart from differences in the apertures, these grains exhibit the same morphological features and micro-rugulate ornamentation with respect to *Vitis vinifera* displayed by standard palynological references [6, 11, 25, 50, 52, 60–62]. However, they differ from the typical morphology of the pollen grains from the non-functioning stamens of female flowers [63], which are aporate and acolpate, but with the same ornamentation of pollen from male flowers [57–59, 64–68].

**Plant tissues.** Remains of plant tissues trapped in or attached to the resin of SFC2 were found during microscopic observation (Fig 4A). By comparison with modern *Vitis* flower observations after dissection of the stamen (Fig 4B), we assigned them to the filament of *Vitis* stamen which connects the anther to the pedicel.

Radial and transversal sections of charred woods were recovered from the pitch of SFC2 and SFC5. Diagnostic features for the identification were: presence of resin canals (Fig 4C), uniseriate rays, and large fenestriform pits in cross-fields. Based on these characters, the wood fragments were identified as *Pinus* group *sylvestris*, including *P. mugo*, *P. nigra* and *P. sylvestris*, whose wood anatomies are undistinguishable from each other with microscopic tools [69, 70].

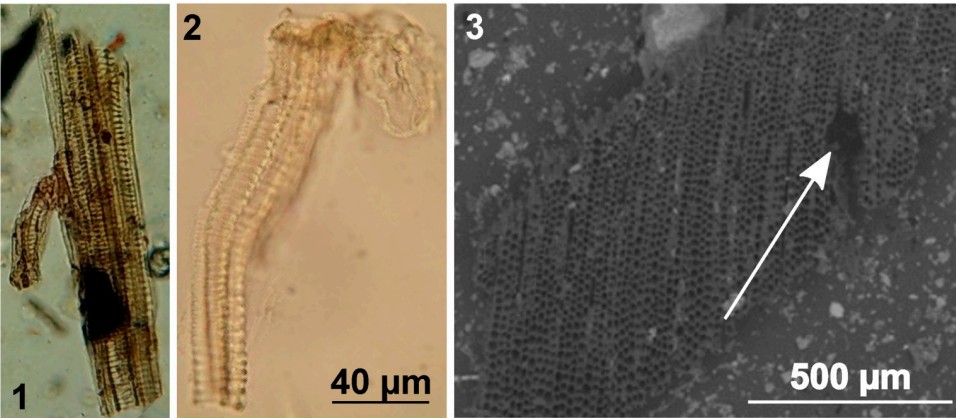

**Fig 4.** Microscopic observation of (A) archaeological plant tissues trapped in the resin of SFC1; (B) filament from the stamen of a modern wild *Vitis vinifera* flower, and (C) ESEM observation of a transverse section of charred *Pinus* wood trapped in the SFC2 pitch. The white arrow indicates the resin canal.

## Discussion

We hereby combined chromatographic tools with archaeobotanical approaches to reach a better understanding of the coating and content of the amphorae and of their use.

### Coating of the amphorae

Chemical and archaeobotanical outcomes frame the use of Pinaceae wood tar to coat the amphorae, also frequently reported in the literature [71, 72]. Aromatic hydrocarbons such as norabietatrienes, retene and simonellite characterize an intensive heating under anaerobic environment [73, 74]. Interestingly, resin is often used to flavor wines, additionally to its bactericide and waterproofing effects [75]. Hostetter *et al.* [76] notably evidenced the resinated wine *vinum picatum* described by Pliny the Elder thanks to an accumulation of resin reported in Etruscan wine cauldrons.

Besides the prominent representation of *Pinus* pollen (sometimes in lumps) that accounts for almost one third (SFC 1 and SFC5) to two thirds (SFC2) of the total grains, the hypothesis of wood pyrolysis is substantiated by the presence of pine charcoal and DHAM markers. Indeed, the DHAM compound is obtained via methanolysis during the distillation of wood: the methanol contained in wood bark esterifies DHA molecules from the diterpenic resin when heated together at very high temperatures [77]. Doubtlessly, wood was consumed during resin production and pollen grains were attached to the hot resin, since pollen resists high temperatures very well [78]. The presence of both pollen and charcoal allowed a better understanding regarding the pitch origin, which is impossible to reach through organic residue analyses alone.

Despite the identification of *Pinus* to the species level is not possible, the botanical assignment to highland pine species (*P. mugo*, *nigra* or *sylvestris)* is strengthened by Pliny, whose *Naturalis Historia* stated that fire-extracted pitch from mountainous species logs of *P. mugo* (namely "taeda") is resin-richer, and notwithstanding its restricted spatial distribution, highland species were abundantly manufactured (Pliny, *N. H.* XIV, 9, 17, 21, 22) [79]. The current distribution of *P. mugo* is very restricted in central Italy, while it is commonly found in alpine environments, where also *P. sylvestris* is widely distributed [80]. In contrast, *P. nigra* is uncommon on the Alps, and sparse in the central and northern Apennines, while the subspecies *P. nigra* subsp. *laricio* is present in Calabria, in Sicily (slopes of Mount Etna) and in Corsica. In

Roman times the production of pitch from the mountains of Calabria and Sicily was renowned (Dionysius of Halicarnassus, XX, 6; Pliny, N. H. III; Cicero, *Brutus* XXII, 85–88). In any case, we can exclude a local origin of the pine used for the production of pitch of the studied amphorae.

The dominance of *Quercus* (46% in SFC1 and 29% in SFC5, Table 1) can be explained by environmental abundance in the region of wood tar production [81].

## Content of the amphorae

The amphora typologies of late Greco-Italic and Dressel 1A (SFC1 and SFC2) and Lamboglia 2 (SFC5) have been frequently reported as grape-derivatives containers [6, 39, 71]. Tartaric acid, together with malic acid (although less specific), identified by GC-MS, point out a grape-based content. Fermentation assumptions is enhanced by succinic and glutaric acids. Pyruvic acid resulting from spontaneous malolactic fermentation refers to wine content [82]. For SFC1, syringic acid present in the second extract, despite its absence in the first extract, originates from malvidin oxidation, thus ruling out potential contamination from free extractible origin [21, 28]. Red (for SFC1) and white winemaking processes (for SFC2 and SFC5) are therefore brought to light. Although the amount of tartaric acid is remarkably higher in grape bunches than in other edible products, its use as a reliable grape biomarker must be confirmed by other evidence [25, 31]. Thus, macroremains of plant tissues recovered from the pitch and identified as part of the filament of *Vitis* flower in samples SFC1 and SFC5 bring this needed evidence of grape derivatives content.

Regarding microremains, although the tricolpate pollen type (Fig 3) does not exhibit any apparent pore, attribution to *Vitis* is straightforwardly demonstrated by the identical aporate 3-zonocolpate grains observed from wild vines (Fig 3C and 3D) and from the Rignano Flaminio fossil sediments from the same Lazio region (Fig 3A and 3B) [48]. The micro-rugulate ornamentation evidenced by SEM observation of Rignano Flaminio sediments and from the surface of wild fruits from Tivoli (S1 Fig) is in accordance with the literature for *Vitaceae* pollen [52]. Nevertheless, attribution to *Vitis vinifera* subsp. *vinifera* is highly questionable since grapevines were surely not domesticated in the Middle Pleistocene [83, 84]. Furthermore, the fossil impression of a grapevine leaf, identified as *V. sylvestris*, coupled with pollen grains, reported from another fossil site 16 km away from Rignano Flaminio, confirms the presence of wild grapevines in this area during the Middle Pleistocene [85]. *Vitis vinifera* subsp. *sylvestris* is well represented in Italy, especially along the Tyrrhenian coast [86, 87]. Although its survival is highly threatened [88], wild grapevines are still present also in southern Latium, close to San Felice Circeo [86, 89].

As far as we know, this is the first time tricolpate aporate *Vitis vinifera* is found in Roman amphorae, although inaperturate (aporate and acolpate) *Vitis* grains were recently retrieved from the Middle Bronze Age site of Terramara di Poviglio [59]. Interestingly, some morphological abnormalities of *Vitis* grains displaying one, two or four pores have also been published in modern cultivars [90].

To truly assume a grape derivative content in the amphorae, the presence of aporate grape pollen in the pitch shall be explained. Three hypotheses are discussed.

**Beverages produced from on-going domestication *Vitis* cultivars.** A first possible answer regards the sterility of the grapevine used. The presence of aporate pollen grains on *Vitis* fruits from wild plants near Tivoli demonstrates the permanence of *Vitis* pollen from female flowers on the fruit surface, despite its development (Fig 3C and 3D). Pollen, subsisting upon time, weather and environmental circumstances, may be plucked with the fruits and remain even in the fermented beverage. Since beverages were not filtered at this time, pollen

could remain in the amphorae attached to the pitch and bring evidence of the ancient content [2, 9, 10]. The pollen we observed might be sterile as suggested by the strikingly thick exine observed, reaching up to 5 μm (Fig 3I and 3J) whereas tricolporate pollen rather exhibits a "thin to fairly thin" wall [52]. Although cytogenetic characteristics are preserved during the grain development, Caporali *et al.* [68] explained *Vitis* pollen sterility by morphological germination inhibition caused by wall structure abnormalities: during the grain formation, the pollen surface is covered by structural nutrients contained in the exine [68]. An excess of exine cover during pollen hydration may turn a mechanical hurdle into sterility [58]. Physiological and cytological functions are nevertheless maintained, and pollen grains can disperse once released by the anthers. The absence of germinative pores causes grain sterility by avoiding pollen tube development even though pollen grains are viable [57, 65].

Pollen sterility is strictly related to wild features. At early stages, flowers of *Vitis vinifera* are all hermaphrodite, until they may become unisexual due to the abortion of one reproductive organ [68, 91, 92]. Sex determining genes therefore divide flowers into functional "male" (or staminate) and "female" (or pistillate), showing up fertile pollen (and rudimentary pistil) or functional pistil (and sterile pollen), respectively [63, 93]. Pollination is achieved through the intermediary of a fertile pollen coming from either a functional male (*V. vinifera sylvestris*) or hermaphrodite plant (*V. vinifera vinifera*).

Pollen sterility also involves dioecy, which was lost during the domestication process [84, 94]. Grapevine domestication targets the ensemble of "genotypic, phenotypic, plastic and contextual impacts that can be used as markers of evolving domesticatory relationships" [95]. Asides the increase of sugar content in the fruits, berry sizes or changes in pips morphology, more factual definitions point to the shift from dioecy to hermaphrodism [84]. The reverting to hermaphrodism is assumed to have occurred through a rare event of male and female haplotypes recombination [59, 94]. Nonetheless, domestication and hermaphrodism have to be clearly separated from each other, since they differently relate to cultivation. Although cultivated *V. vinifera* are thought to have been domesticated from their wild *V. vinifera sylvestris* ancestors [96], not all the cultivated plants were necessarily hermaphrodite at the beginning of domestication, which was a long and multi-located process [19, 97, 98]. As observed from wild cereals, grapevine cultivation for food consumption is presumed to have started long before its domestication [25, 99, 100]. One point remains certain: the switch to hermaphroditism grandly facilitates the fruit production, turning grapevines into self-pollinating plants, with entomophilous and anemophilous cross-pollination [84, 101].

Additional evidence for cultivation is based on a considerable progress of statistic and modeling morphometric tools applied to pips and the emerging field of ancient DNA [19, 23, 83, 102–104]. SSR and SNP markers allowed genotype classification into cultivated and wild types [105, 106]. However, despite the important genetic dissimilarity reported between *V. vinifera* and *V. sylvestris*, a remaining presence of wild characters cannot be ruled out [97, 105, 107].

Historical and archaeological evidence supports the use of wild grapes at the same time of cultivated grapes. In his *Naturalis Historia*, Pliny repeatedly reported the use of *V. sylvestris* grapes, wood and leaves in addition to cultivated grapevines (Pliny, *N. H.* XIV). In the Middle to Late Bronze Age site of Santa Rosa di Poviglio in the Po Plain, tricolporate *Vitis* pollen was abundantly recovered, up to 18% [108]. The re-examination of the archaeological sediments highlighted 15 inaperturate *Vitis* pollen grains that demonstrate the use of *V. sylvestris* [26, 59]. Although present in limited quantities (7.7% of the total grape pips), Mariotti Lippi *et al.* [23, 109, 110] identified grape pips as wild morphotypes in Tuscany, belonging to Middle Bronze Age and Etruscan-Roman archaeological contexts. Castellano [111] pointed out the presence of *V. sylvestris* pollen in honey dated to the Iron Age in northern Italy, suggesting that bees fed

on nectar of pre-domesticated or early-domesticated varieties of *V. vinifera*. Bouby [19] highlighted an intermediate form of Roman grape pips, between "highly cultivated" and "primitive cultivars" from wild ancestors, in the South of France. In synthesis, while wild male flowers produce tricolporated pollen similar to modern hermaphrodite cultivars species, modern wild female flower rather produce aporate pollen grains [59].

The *Vitis* pollen retrieved from the Roman amphorae of San Felice Circeo may therefore represent an intermediary stage of domestication, being characterized by thick exine, absence of germinative pores and presence of colpi. This intermediate morphology recalls the morphological variety observed in stamens and pistils of *Vitis* flowers (functional female flowers present either erect but crinkled stamens, or semi- as well as entirely reflexed stamens) [85]. Advanced archaeopalynological analyses are needed for a better understanding of the grapevine evolution from wild, through intermediate to cultivated forms [26, 108, 112]. This field of research offers new eyes for an innovative archaeological interpretation of the data [99, 113, 114].

**Beverages produced from dioecious *Vitis* cultivars.** A second hypothesis to explain the presence of aporate *Vitis* pollen in the Roman amphorae of Circeo concerns the use of dioecious cultivars to produce fermented grape derivatives. Several modern cultivars, such as 'Loureiro', 'Moscato rosa' or 'Blatina', have been documented to compose with a dioecious mating system [64, 115–117]. Some of them, like 'Picolit' and 'Lambrusco di Sorbara' are endemic of northern Italy [59, 65]. Since the plants of these refined wines cannot self-fertilize due to infertile pollen from functionally female flowers, grapevine productivity is consequently limited and wine prices may be high [118]. Dioecious varieties used for wine production are indeed uncommon, but they can nevertheless be found in various regions and are not specifically local. Our study of modern pollen from different sites near Rome, showing tricolpate aporate grains, supported by Middle Pleistocene records with the same pollen morphology confirms the existence of natural populations of *Vitis* with the aporate pollen type. Such wild grapes, that could be rather common before the spread of *Phylloxera* in Europe in the 19th century [84] could have been used to produce the wine content of the Roman amphorae from San Felice Circeo. Wild grapes could be used to produce a local table wine destined to common use but also a refined wine, like the 'Picolit', meant for a refined cuisine.

**Beverages produced from wildflowers of *Vitis* vinifera.** The third hypothesis refers to the nature of the content. As previously described, chemical markers evidenced fermented grape derivatives, which can consist of either wine, vinegar or other beverages, such as the cooked wine '*defrutum*' or aged sweetened wine '*mulsum*' [6, 119, 120]. Some Roman recipes reported in *De re coquinaria V.II.9*, translated and abridged by Feldman [121], attested of the use of grape wine in traditional cooking made out of roses, spices and fish sauces of *liquamen* or *garum*. Unfortunately, in the absence of *tituli picti*, i.e., a commercial inscription on the surface of the amphorae, the possibilities of wine derivatives remain hypothetical since no chemical biomarker is able to distinguish wine from other traditional grape fermented recipes [122, 123]. Relying on the botanical evidence, we hereby consider the possibility of *oenanthium*, a flavored wine famous for its medicinal properties. Following Pliny recipe "w*ith the wild grapevine one makes what is called* oenanthia: *one macerates two pounds of wild vine flower in a cadus (30 or 40 liters) of must, one decants after thirty days [. . .]. These grapes, shortly after flowering, are a remedy of singular virtue to temper the heat of the body in diseases*" (Pliny, *N. H.* XIV, 18). Besides attesting the use of wild flowers at Roman times, Pliny gives an interesting justification of *Vitis* pollen in jars through the presence of flowers. More importantly, we have also found stamen filaments (Fig 4), which support the presence of flowers in the content of the amphorae. Moreover, a few non-arboreal pollen types may account for medicinal aromatization, specifically in SFC5 where *Artemisia* reached 8.3% of the total grains and SFC1 where

*Myrtus* has been identified. Such plants are recognized for their medicinal benefits [124, 125]. Broadly speaking, there is abundant ethnopharmacological evidence for the common use of herbal concoctions in alcoholic or fermented beverages [126]. The use of *Artemisia* for anti-cancer activity has been reported in Ancient Egyptian herbal wines from Abydos potsherd [127, 128]. *Myrtus communis* was recognized as a therapeutic drug in ancient Greece [129] and used as herbal additives in early medieval beers [130]. Nonetheless, the limited pollen representation of *Myrtus* (less than 2%) can also refer to the surrounding vegetation.

In synthesis, our results suggest the use of autochthonous grapevines either cultivated, such as the modern 'Picolit', or wild, as demonstrated by the similarities with *Vitis* pollen from indigenous wild plants, as well as the possible use of grapevines at intermediate stages of domestication. Medicinal wine is another possibility that would explain the presence of *Vitis* stamens, and of *Artemisia* and *Myrtus* pollen, whose plants are used as flavoring. Likewise, *Pinus*, besides ensuring the waterproofing of the amphora, would have flavored the beverage due to its aromatic character. Indeed, herbal wines such as *vinum absinthianum* or *picatum* were common at that time.

## Conclusion

The analyses of the pitch contained in three Roman amphorae from San Felice Circeo illustrates the benefits of applying a multidisciplinary approach. The combined evidence of amphorae typologies, Pliny's testimony regarding the use of *V. sylvestris*, previous archaeobotanical finds indicating the archaeological use of wild grapes, the chromatographic outcomes, and the morphology of *Vitis* pollen and tissues led to new archaeological and anthropological interpretations.

The identification of *Pinus* group *sylvestris* used to produce wood tar for waterproofing, matching the methyl ester diterpenic chemical markers characterized by GC-MS, indicates a non-local origin of the wood tar, as also suggested by ancient historical sources, reporting Calabria and Sicily as important production areas for pitch.

Aside from confirming the usage of Lamboglia 2 as wine containers, chemical analyses highlight the usage of both red and white wines.

The observation of aporate pollen of *Vitis*, compared with different types of fossil and modern wild grapevines, suggests the use of autochthonous grapevines, either wild or cultivated, without excluding a possible intermediary stage of domestication of cultivars still bearing *V. sylvestris* features. It is also possible to conjecture the archaeological presence of a medicinal grape beverage made as an infusion of wild *Vitis* flowers in the must, reported by Pliny as *oenanthium*. However, this hypothesis contrasts with the diverse typology of the amphorae involved in the pitch analyses.

*Vitis* pollen appears to be a fruitful anthropological indicator of ancient habits by opening a field of archaeological assumptions hitherto inaccessible, such as the inclusion of Roman grapevines into the long process of domestication. Within the long-standing question of distinguishing wild and cultivated grapes from past archives, the archaeopalynological study of *Vitis* may bring new evidence to define the timing and modes of grapevine cultivation.

Since false chemical positive must be tackled by external controls, we provided a straightforward methodology that brought independent evidence of grape derivatives in Roman wine amphorae, based on chromatographical and archaeobotanical tools, allowing to suggest a history beyond the artefacts that could not be identified by single analytical techniques.

## Supporting information

**S1 Fig.** ESEM pictures of *Vitis vinifera* pollen grains recovered from (A) grapefruits from wild grapes in Tivoli and (B) Fossil sediment from Rignano Flaminio. Pollen grains are tricolpate, ranging from 18–27 μm, with micro-rugulate ornamentation. Unlike SEM references for

Vitaceae [49], no porus was observed along the colpi.
(DOCX)

## Acknowledgments

The authors of this work are grateful to Fabrizio Michelangeli for his great support in pollen identification.

## Author Contributions

**Conceptualization:** Louise Chassouant, Alessandra Celant, Chiara Delpino, Cathy Vieilles-cazes, Carole Mathe, Donatella Magri.

**Formal analysis:** Louise Chassouant, Alessandra Celant, Federico Di Rita, Donatella Magri.

**Funding acquisition:** Cathy Vieillescazes, Carole Mathe, Donatella Magri.

**Investigation:** Louise Chassouant, Alessandra Celant, Chiara Delpino, Federico Di Rita, Donatella Magri.

**Methodology:** Louise Chassouant.

**Project administration:** Carole Mathe, Donatella Magri.

**Resources:** Alessandra Celant, Chiara Delpino, Donatella Magri.

**Visualization:** Louise Chassouant.

**Writing – original draft:** Louise Chassouant, Alessandra Celant, Chiara Delpino, Donatella Magri.

**Writing – review & editing:** Louise Chassouant, Alessandra Celant, Chiara Delpino, Federico Di Rita, Cathy Vieillescazes, Carole Mathe, Donatella Magri.

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
