## [Decision Letter · Decision Letter 0]

10 Nov 2021

PONE-D-21-28574Archaeobotanical and chemical investigations on wine amphorae from San Felice Circeo (Italy) shed light on grape beverages at the Roman timePLOS ONE

Dear Dr. Chassouant,

Thank you for submitting your manuscript to PLOS ONE. After careful consideration, we feel that it has merit but does not fully meet PLOS ONE’s publication criteria as it currently stands. Therefore, we invite you to submit a revised version of the manuscript that addresses the points raised during the review process.

The manuscript has been revised by a reviewer that raised a series of critical comments. Please revise the manuscript that will be sent to additionall reviewers. ==============================

We look forward to receiving your revised manuscript.

Kind regards,

Raffaella Balestrini

Academic Editor

PLOS ONE

Journal Requirements:

"This research was financially supported by Marie Skłodowska-Curie Innovative Training Network, The ED-ARCHMAT European Joint Doctorate, H2020-MSCA-ITN-EJD ED-ARCHMAT Joint Doctorate (Project ESR9, grant agreement no 766311). The authors of this work are grateful to Fabrizio Michelangeli for his great support in pollen identification."

"This research was financially supported by Marie Skłodowska-Curie Innovative Training Network, The ED-ARCHMAT European Joint Doctorate, H2020-MSCA-ITN-EJD ED-ARCHMAT Joint Doctorate (LC Project ESR9, grant agreement no 766311). The funders had no role in study design, data collection and analysis, decision to publish, or preparation of the manuscript"

Reviewers' comments:

Reviewer's Responses to Questions

**Comments to the Author**

1. Is the manuscript technically sound, and do the data support the conclusions?

Reviewer #1: Partly

2. Has the statistical analysis been performed appropriately and rigorously? 

Reviewer #1: Yes

3. Have the authors made all data underlying the findings in their manuscript fully available?

Reviewer #1: Yes

4. Is the manuscript presented in an intelligible fashion and written in standard English?

Reviewer #1: Yes

5. Review Comments to the Author

Reviewer #1: The manuscript presents a study of Roman wine amphorae combining chemical palaeobotanical analyses. It provides interesting insights and leads to original hypotheses about the grape juice beverages that may have been contained in the amphorae and about the vines that produced them.

My advice is to publish the article with minor changes. The article could be shortened a bit and presented in a more concise form. It contains a large number of bibliographical references. The « Materials and Methods » section is lengthy and contains details that do not seem necessary, in particular in the « Archaeological context » and « Solvents » sections. The introduction is also a bit long and mainly takes the form of a historiographical presentation of the various chemical and palynological analyses of amphora contents carried out so far. For the sake of the article it would be better to have a slightly more concise introduction, not focusing so much on the methodological angle but highlighting the questions addressed by the study of the type of material and by this specific study.

One of the most innovative aspects of the article certainly concerns the observations on Vitis pollens and the hypotheses that are drawn from them on the nature and origin of the vines used for the production of the beverages contained in the amphorae.

However this aspect of the article deserves to be clarified.

In section « Pollen » (202) it is a bit difficult to understand which pollen morphotypes can be encountered in modern vines and which are closer to the archaeologicla pollen grains. It would be good to discribe briefly the morphotypes typical of modern wild male flowers, modern wild female flowers and modern hermaphrodite domesticated varieties (female varieties if known).

As it stands I am not convinced at all by your second hypothesis, the use of indigenous cultivars. Contrary to what you say, I do not think that the data you present strongly support this hypothesis. You list several indigenous female varieties. Such varieties are indeed in the minority, but they can nevertheless be found in various regions and are not specifically local. The resemblance to local Pleistocene pollens does not seem to me to be a valid argument either; or Pleistocene or early Holocene pollens from other regions should be shown to have a different morphology. If I have misunderstood your argument, I apologise and please clarify your thinking. Otherwise I think that this second hypothesis should be abandoned.

- Minor comments.

- Table 1 ; I don’t see how you could have 1.6% Erica from a total of 31 pollen grains. Please check all values in Table 1.

- 305. Please replace « in this area since at least » by « in this area during the Middle Pleistocene ». Or else provide evidence that wild grapevine was recorded since the Middle Pleistocene.

- 384-407. Third hypothesis. I would guess that Pliny refers to male wild flowers. Can you confirm that or do you have more specific information on that matter?

6. PLOS authors have the option to publish the peer review history of their article (what does this mean?). If published, this will include your full peer review and any attached files.

Reviewer #1: No

---

## [Author Response · Author response to Decision Letter 0]

10 Feb 2022

Additionally, we would like to thank the reviewer for the important comments he reported in the manuscript. In red (in the "Response to reviewer" document), we answered and detailed the modifications we proposed.

Reviewer #1: The manuscript presents a study of Roman wine amphorae combining chemical palaeobotanical analyses. It provides interesting insights and leads to original hypotheses about the grape juice beverages that may have been contained in the amphorae and about the vines that produced them. My advice is to publish the article with minor changes. The article could be shortened a bit and presented in a more concise form. It contains a large number of bibliographical references. 

 Considering your remark, this is true that the publication includes a large number of bibliographical references. However, the present paper aims at combining 3 different disciplines (archaeology, botanical analysis and organic residue / chemical analysis). The number of citations required is therefore elevated to provide a concrete and well-documented work. 

The « Materials and Methods » section is lengthy and contains details that do not seem necessary, in particular in the « Archaeological context » and « Solvents » sections.

 The “Solvents” section was entirely removed.

 The sections “Sample preparation for chromatographic analyses” and “Gas chromatography – Mass Spectrometry” were combined. The complete extractive protocol had been recently published; hence the citation was uploaded.

 The “Archaeological context” section remains fundamental for the well-understanding of the article, specially for the archaeological consideration of the site to put in perspective the archaeobotanical results. Moreover, the archaeological details consigned in the introduction are helpful to understand the results obtained. Therefore, it seems complicated to us to shorten this section without detracting from the overall understanding of the article and the archaeobotanical results.

The introduction is also a bit long and mainly takes the form of a historiographical presentation of the various chemical and palynological analyses of amphora contents carried out so far. For the sake of the article it would be better to have a slightly more concise introduction, not focusing so much on the methodological angle but highlighting the questions addressed by the study of the type of material and by this specific study.

 Considering your comment, the introduction was slightly shortened. However, the introduction is less than 160 words and aimed at detailing all of the 3 disciplines considered in the research article. A particular interest was given to highlight the interdisciplinary and innovative character from a methodological point of view since such combined studies have been rarely conducted. In these conditions, it remains complicated to shorten more the introduction without impacting its integrity. 

Following the PlosOne journal recommendation for the introduction, we provided the background context with a methodological angle before naming the purpose and the significance of the study. We carefully developed the problematic regarding the urging need of interdisciplinary methods to understand archaeological objects to prevent overinterpretation or misinterpretation raised by single method results. 

One of the most innovative aspects of the article certainly concerns the observations on Vitis pollens and the hypotheses that are drawn from them on the nature and origin of the vines used for the production of the beverages contained in the amphorae. However this aspect of the article deserves to be clarified. In section « Pollen » (202) it is a bit difficult to understand which pollen morphotypes can be encountered in modern vines and which are closer to the archaeologicla pollen grains. It would be good to discribe briefly the morphotypes typical of modern wild male flowers, modern wild female flowers and modern hermaphrodite domesticated varieties (female varieties if known).

 A sentence was added (L. 367) to summarize the different morphologies encountered.

As it stands I am not convinced at all by your second hypothesis, the use of indigenous cultivars. Contrary to what you say, I do not think that the data you present strongly support this hypothesis. You list several indigenous female varieties. Such varieties are indeed in the minority, but they can nevertheless be found in various regions and are not specifically local. 

 We thank the reviewer for this comment which gives us the opportunity to better explain the second hypothesis. We added a sentence as suggested (L. 384).

The resemblance to local Pleistocene pollens does not seem to me to be a valid argument either; or Pleistocene or early Holocene pollens from other regions should be shown to have a different morphology. If I have misunderstood your argument, I apologise and please clarify your thinking. Otherwise I think that this second hypothesis should be abandoned.

 Reading again our second hypothesis, it is true that it was not explained clearly enough. For this reason, we made changes throughout the paragraph to mitigate the interpretation. It remains important for us to keep this hypothesis as one of the possible interpretations to the understanding of the tricolpate pollen in order to leave open a discussion on the subject since this pollen has not been reported in these conditions.

---

## [Decision Letter · Decision Letter 1]

4 Apr 2022

Archaeobotanical and chemical investigations on wine amphorae from San Felice Circeo (Italy) shed light on grape beverages at the Roman time

PONE-D-21-28574R1

Dear Dr. Chassouant,

We’re pleased to inform you that your manuscript has been judged scientifically suitable for publication and will be formally accepted for publication once it meets all outstanding technical requirements.

Kind regards,

Raffaella Balestrini

Academic Editor

PLOS ONE

Additional Editor Comments (optional):

Reviewers' comments:

Reviewer's Responses to Questions

**Comments to the Author**

1. If the authors have adequately addressed your comments raised in a previous round of review and you feel that this manuscript is now acceptable for publication, you may indicate that here to bypass the “Comments to the Author” section, enter your conflict of interest statement in the “Confidential to Editor” section, and submit your "Accept" recommendation.

Reviewer #2: All comments have been addressed

2. Is the manuscript technically sound, and do the data support the conclusions?

Reviewer #2: Yes

3. Has the statistical analysis been performed appropriately and rigorously? 

Reviewer #2: N/A

4. Have the authors made all data underlying the findings in their manuscript fully available?

Reviewer #2: Yes

5. Is the manuscript presented in an intelligible fashion and written in standard English?

Reviewer #2: Yes

6. Review Comments to the Author

Reviewer #2: I was invited to read the revised version of this very interesting and interdisciplinary paper (I was not a reviewer of the first version). The research adds another piece of evidence to the history of Vitis use, the discussion is in-depth, complete and takes into consideration the complexity of the disciplines, archaeological context, and botanical features. Conclusions agree with the most recent scientific evidence and are convincing.

The paper can be accepted in the present form, or a bit improved by the following minor remarks.

Below, I only ask for adding details about the Vitis morphology reported in the text, to avoid ambiguity.

Line 137 = it is not clear in this first sentence what subspecies was sampled.

Lines 203 and following: in my personal experience, all pollen from functionally female flowers of Vitis vinifera (both domesticated or wild subspecies) were observed inaperturate, that means no pores nor colpi. Maybe, this could be explained at line 207: pollen grains from female flowers of wild grapevine were observed as inaperturate pollen grains.

Therefore, it is new the form found in these cultivars of aporate 3-zonocolpate pollen grains produced by female flowers.

Citation 57 = add the journal?

Line 210 = “the aporate morphology of Vitis vinifera was also found in pollen grains from sediments belonging to the Middle Pleistocene sediments of Rignano Flaminio” = do you mean ‘aporate’ as inaperturate rounded pollen (as in my experience of wild subspecies) or ‘aporate 3-colpate’ ?

Line 275 = the identical aporate grains are tricolpate?

The chapter ‘content of the amphorae’ is very interesting and articulated: maybe of interest to subdivide it into subchapters, especially to outline better the presence of most information from historical sources.

Anna Maria Mercuri

7. PLOS authors have the option to publish the peer review history of their article (what does this mean?). If published, this will include your full peer review and any attached files.

Reviewer #2: **Yes: **Anna Maria Mercuri

---

## [Editor Report · Acceptance letter]

12 Apr 2022

PONE-D-21-28574R1 

Archaeobotanical and chemical investigations on wine amphorae from San Felice Circeo (Italy) shed light on grape beverages at the Roman time 

Dear Dr. Chassouant:

I'm pleased to inform you that your manuscript has been deemed suitable for publication in PLOS ONE. Congratulations! Your manuscript is now with our production department. 

Kind regards, 

on behalf of

Dr Raffaella Balestrini 

Academic Editor

PLOS ONE